# Does Energy Restriction and Loss of Body Fat Account for the Effect of Intermittent Fasting on Cognitive Function?

**DOI:** 10.3390/nu17152407

**Published:** 2025-07-23

**Authors:** Fulvia Draicchio, Kathleen V. Axen

**Affiliations:** Department of Health and Nutrition Sciences, Brooklyn College of the City University of New York, Brooklyn, NY 11215, USA

**Keywords:** intermittent fasting (IF), Time-Restricted eating (TRE), caloric restriction (CR), cognitive function, brain health, visceral adiposity, insulin resistance, weight loss, body fat, Inflammation

## Abstract

Intermittent fasting (IF) and caloric restriction (CR) have garnered attention for their potential to enhance cognitive function, particularly in aging and metabolically compromised populations. This narrative review critically examines whether the cognitive benefits of IF are attributable to its specific fasting patterns or are instead mediated by the production of weight loss, fat loss, and improvements in insulin sensitivity. Evidence from human trials suggests that reductions in body weight, especially visceral adiposity, and improvements in insulin resistance are key factors linked to enhanced cognitive performance. Comparisons between the results of IF and CR trials show comparable cognitive outcomes, supporting the idea that negative energy balance, rather than fasting or altered eating patterns, drive these effects. However, further investigation of specific types of IF patterns, as well as possible cognitive domains to be affected, may identify mechanisms through which IF can yield benefits on neurocognitive health beyond those of loss of body fat and its accompanying inflammatory state.

## 1. Introduction

The brain’s metabolic state and cognitive function are influenced by the physiological function and nutritional state of the body as a whole. A number of cross-sectional studies have linked obesity and insulin resistance with detrimental effects on cognitive performance [1,2,3]. Given that some impairment of cognitive performance often precedes, by years, functional and neurodegenerative changes, identification of precautionary measures for at-risk individuals may be important for the prevention or mitigation of impairment. Based on these premises, recent studies have tested the effects of reducing obesity and insulin resistance on indices of cognitive function, through restriction of energy intake [4,5]. This restriction may involve a general reduction in energy intake, or a change in the timing and pattern of energy intake, or both.

Loss of body weight has been associated with improved cognitive function in some human studies [4,6,7]; several studies have indicated that it is the loss of body fat that is relevant to this effect. Increased waist circumference (WC), a clinical marker of central body fat (visceral + subcutaneous), is more closely associated with elevated inflammatory markers and cognitive deficits than Body Mass Index (BMI kg/m^2^) alone [8,9]. Longitudinal research shows that individuals with higher WC in midlife exhibit a greater risk of late-life dementia and Alzheimer’s disease [10]. Uchida et al. followed a cohort of over 2500 Japanese adults aged 40–65 at baseline for more than 15 years and found that those in the highest quartile of waist circumference had nearly double the risk of developing dementia compared to those in the lowest quartile, even after adjusting for confounding variables such as age, sex, diabetes status, and physical activity [11]. Structural neuroimaging studies showed that central adiposity correlated with decreased gray matter in the hippocampus and prefrontal cortex, two regions that subserve memory and executive function [12]. In the study by Ozato et al. [13], brain MRI data from 1200 middle-aged adults revealed that higher visceral fat area (measured by abdominal CT) was significantly associated with reduced cortical thickness and volume in frontal and temporal regions of the brain, even after adjusting for BMI, blood pressure, and lipid profiles. These findings suggest a relationship between visceral fat content and brain structure.

Dietary interventions such as intermittent fasting (IF) and caloric restriction (CR) have garnered significant attention for their potential to mitigate impairments in cognitive function [4,14,15]. Intermittent fasting, an umbrella term for various patterns that alternate between periods of eating and extended fasting, is distinct from traditional caloric restriction (CR) in that it emphasizes the timing and rhythm of food intake. Here, “rhythm” refers to the consistent recurrence of fasting-eating cycles (e.g., daily or weekly), which is believed to evoke physiological states mimicking those experienced by humans during ancestral periods of food scarcity. However, it is not known whether the pattern of food intake of IF is an essential part of the mechanism for improving cognitive function, or whether the timing and rhythm only serve to actualize and sustain the reduced energy intake. Furthermore, it is not known whether the benefits of energy restriction—loss of body fat and reduction in insulin resistance—are themselves involved in ameliorating cognitive function, regardless of how energy intake is reduced.

The major questions that the current review addresses, with respect to these approaches to protecting or improving cognitive function, are as follows:(1)Is negative energy balance, regardless of how it is produced, responsible for the effects of these interventions on cognitive performance?(2)Is reduction in obesity (loss of body fat), or improvement in insulin resistance a necessary component in the effect of IF or CR on cognitive performance?

The first question has been studied in various animal models by comparing the effectiveness of different means of weight loss [16]. However, rodent eating patterns differ from those in humans in many ways, including nocturnal eating, much shorter life-span (fraction of life represented by a 16 h fast) and responses to meal vs. ad libitum feeding regimens; these differences lessen the applicability of findings to humans. A recent review by O’Leary et al. [4] concluded that there is ample evidence that various methods of restricting energy intake can be able to influence cognitive function, but that the cognitive domains may vary with type and degree of energy restriction. The current narrative review will compare human trials using various forms of IF with those using CR, not to investigate the relative efficacy of the two methods in preserving or improving cognitive function, but to examine the influence of the eating patterns imposed by IF separately from IF’s effects on body weight or body fat loss. Such a comparison can inform application of dietary protocols to affect cognitive function and may advance exploration of the mechanisms through which energy restriction influences the function of the brain.

The second question will focus on the restriction of energy intake by IF and CR in those studies that provide outcomes related to body composition and insulin resistance, in addition to cognitive performance. The review will consider whether changes in obesity-related measures are necessary for energy restriction to exert its effects. The review will not address the use of protocols that test different diet compositions or which are used in treating neurogenerative diseases, such as Alzheimer’s Disease.

Our research question concerns the Population of adults with overweight or obesity. The Intervention is dietary change. The Comparison is of two modes of dietary change, Intermittent Fasting and Caloric Restriction, for the purpose of isolating loss of body weight or body fat as the mediator of the Outcome, which is a change in cognitive function.

Intermittent Fasting (IF) will be used as an umbrella term for the following:Time-restricted eating (TRE): daily fasting intervals (e.g., 14–18 h) followed by an eating window (typically 6–10 h), with no intentional caloric reduction [17].Alternate-day fasting (ADF): alternating 24 h fasting periods (complete fast or low-calorie) with days of ad libitum eating [18,19].Alternate-day modified fasting (ADMF): a version of ADF where fasting days include a low energy intake (~25–30% of daily needs) [20]The 5:2 diet (fasting mimicking diet): two non-consecutive days per week of caloric restriction (~500–700 kcal), with ad libitum eating on other days [21]

In contrast to IF, CR involves a chronic daily reduction in energy intake (typically 20–40%) without prolonged fasting other than during sleep. The nutrient content of the diets should be adequate to maintain health.

## 2. Methods

An extensive literature search was performed using the electronic database, PubMed, with the generation of ancillary material on Google Scholar. Searches were conducted between January and May 2025, covering peer-reviewed publications from January 1995 through April 2025. The search strategy combined Medical Subject Headings (MeSH) and free-text keywords using Boolean operators. The following keywords and combinations were used:“intermittent fasting” OR “TRE” OR “ADF” OR “ADMF” OR “5:2 diet” OR “CR” OR “weight loss” AND “cognition” OR “cognitive function” OR “executive function” OR “memory” OR “attention”.“caloric restriction” OR “energy reduction” OR “negative energy balance” AND “cognition” OR “brain function” OR “neuroplasticity” OR “BDNF” OR “hippocampus”.“obesity” OR “central obesity” OR “body fat” OR “visceral fat” OR “metabolic syndrome” OR “waist circumference” AND “cognitive impairment” OR “cognitive decline”.

Follow-up literature searches utilized references from the review articles that were generated by the searches. Only articles indexed in peer-reviewed journals and meeting inclusion criteria were considered.

**Inclusion Criteria:** (1) Study design: randomized controlled trials (RCTs), longitudinal studies, cohort studies, and human observational studies. (2) Population: adult men and women. (3) Intervention studies examining IF protocols (e.g., time-restricted eating, alternate-day fasting, 5:2 diet), CR protocols, or bariatric surgery (a means of CR) with clear outcome measures. (4) Outcomes: Measures of cognitive performance, brain function, or neuroprotection, AND weight loss, body fat loss, insulin sensitivity, glycemic control, or inflammatory markers. (5) Language: English.

**Exclusion Criteria:** (1) IF or CR groups only in combination with modifications of dietary macronutrient composition. (2) Focus solely on athletic or underweight populations. (3) Use of diets to treat diagnosed neurodegenerative disorders.

The authors independently screened titles and abstracts, followed by full-text review to assess eligibility based on the criteria above. Disagreements were resolved through consensus.

Eligible studies were analyzed based on population characteristics, experimental design, cognitive and metabolic outcome domains, study duration, and main findings. Systematic reviews and meta-analyses were used to identify knowledge gaps and provide broader context for interpreting primary data, as well as to identify additional relevant research studies. Only studies that provided detailed description of their experimental design, whether pre vs. post, or intervention vs. control, were used. Reporting of anthropometric results, such as change in BMI, body weight, body fat or waist circumference was required. Only statistically significant effects are reported in this review (Figure 1).

## 3. Results

### 3.1. Question 1: Is Negative Energy Balance, Regardless of How It Is Produced, Responsible for the Effects of These Interventions on Cognitive Performance?

#### 3.1.1. Cognitive Effects of Intermittent Fasting

Among the many studies investigating the effects of IF on cognitive function in humans [7], three were found to have protocols that provide data on cognitive function and body composition with either pre vs. post or intervention vs. control comparisons (Table 1).

Anton et al. [21] conducted a 4-week pilot trial evaluating time-restricted feeding (TRE; 16 h fast:8 non-fast) in overweight adults aged 65–75 years (BMI 25–35 kg/m^2^). Despite a significant decrease in body weight and BMI, there was no significant improvement in global cognitive function, measured via the Montreal Cognitive Assessment (MoCA) following IF. However, participants did report subjective enhancements in mental clarity and energy.

Kapogiannis et al. [15] conducted a highly documented, 8-week randomized controlled trial in sedentary older adults with obesity and prediabetes (mean BMI ≈ 32 kg/m^2^). Participants were assigned to either an IF (TRE; 14:10), or a Healthy Living (HL) control diet that did not focus on energy restriction. Statistically significant improvements occurred in both groups in executive function, logical memory and fluency factor, while several other aspects of cognitive function improved only in the IF group. While both groups showed significant losses in body weight, BMI and waist circumference, the effects on body weight and BMI were greater in the IF group. The IF, but not the HL control group, had significant reductions in fasting plasma insulin levels, insulin resistance (HOMA2-IR), HbA1c (an index of chronic glycemic control), and neuronal insulin resistance. The large battery of cognitive tests in this study allowed for greater investigation of possible differences between the two groups.

Ooi et al. [22] obtained cross-sectional data on older adults who self-selected to practice religious long-term intermittent fasting (TRE twice a week; ~14:10) or no fasting. Data on anthropometry, biochemical indices and cognitive function were collected at baseline and after 36 months. The IF group had better scores on the Mini-Mental State Examination (MMSE) and MoCA, suggesting enhanced global cognitive function. This group also exhibited reduced oxidative stress, lower inflammation (C-Reactive Protein, CRP), and increased antioxidant capacity. However, given that the IF group had significantly better indices of health at baseline (lower body weight, blood pressure, plasma insulin levels, lack of smoking), as well as long-time adherence to TRE, the differing habits of the groups confound analysis of the effects of IF. Ooi et al. [23] expanded this work with a mediation analysis, concluding that the cognitive benefits associated with IF were mediated by improved oxidative stress profiles, lower inflammatory markers, and reduced DNA damage.

These investigations of IF showed that significant weight loss per se was not always associated with improvements in the particular indices of cognitive function that were tested, either in Anton et al. [21] or Kapogiannis et al. [15] However, the magnitude of weight loss could be a factor in determining the effect of IF on cognitive performance. The IF group in the study by Kapogiannis et al. lost 4.96 kg, whereas the IF subjects in Anton et al. only lost ~2.25 kg.

#### 3.1.2. Cognitive Effects of Caloric Restriction

Traditional caloric restriction without a structured change in the eating-fasting cycle was used in a by Kretsch et al. (1997) [24] on young women (25–42 years) via a 50% reduction in energy intake for 15 weeks; control subjects maintained their habitual diets (Table 2).

CR subjects lost ~12 kg body weight of which ~10 kg was body fat. Tests of cognitive function were administered to CR subjects at baseline, during CR (weeks 5, 10, 15) and during weight maintenance (weeks 17, 18). Scores on short-term memory tasks and simple reaction time worsened in the CR group and were correlated with the change in body weight. Although this result is counter to that reported in most later reviews, deterioration in cognitive performance under fasting or marked energy restriction had been reported previously [32].

In the study by Witte et al. [25], healthy adults (mean age 60.5 years) underwent a 30% CR intervention for 3 months (initial mean BMI 30 kg/m^2^), resulting in a significant reduction in body weight in the CR group. Compared to the control group (BMI 27 kg/m^2^) which was instructed to maintain habitual intake, CR participants showed significant improvement in verbal memory as measured by the Verbal Learning and Memory Test (VLMT) recognition score. These improvements correlated with reduced fasting insulin and CRP levels, suggesting modulation of insulin sensitivity and inflammation as mediators.

Prehn et al. [14] conducted a 16-week CR intervention (8 weeks @ 800 kcal/d, 4 weeks @ unspecified energy reduction until ≥10% loss of body weight, followed by 4 weeks weight maintenance) in postmenopausal women (mean BMI ≈ 35). Body weight and waist circumference decreased in the CR group, as well as HbA1c. Only CR subjects had significant improvements in recognition memory, delayed recall, processing speed, and executive function (e.g., Stroop task, Trail Making Test B) at week 12, but not after 4 weeks of weight maintenance. These benefits were paired with increased gray matter volume in the hippocampus and inferior frontal gyrus and enhanced hippocampal connectivity, also only at week 12, supporting a potential improvement of neuroplasticity by CR.

Several studies analyzed the effect of continuous 25%CR by non-obese adults, reported from the 24-month randomized control trial, Comprehensive Assessment of Long-term Effects of Reducing Intake of Energy (CALERIE; [33]). This highly controlled and documented study provides extensive information about long-term modest energy restriction in non-obese subjects, as compared with ad libitum eating (control group). After 6 months of CR, no difference was observed between CR and control subjects (each *n* = 12) in tests of memory or attention, despite greater weight loss and fat loss with CR [26,34,35]. At 12 and 24 months of the study, no differences in cognitive function were seen between the CR and control groups; note that actual energy restriction went from ~20% at 6 months to ~10% for the rest of the trial [34]. Reduction in energy intake in these non-obese subjects did not lead to decrements in cognitive performance. Leclerc et al. [27] reported a “slightly positive effect” on spatial working memory (SWM) after 24 months only for the CR group, based on their total SWM errors but not SWM strategy. However, utilizing the same available data set, Silver et al. [28] concluded that there was no difference in effect between the CR and control subjects; only Silver et al. reported that the CR group performed significantly better at baseline than the control group [28].

Another approach to reducing energy intake, without required intermittent fasting, is bariatric surgery. The major form of surgery is Roux-en-Y Gastric Bypass (RYGB). This procedure not only decreases the size of the gastric pouch, thereby tending to reduce the size of each meal, but it also alters the nutrient content of digested food that reaches the lower parts of the intestine, where it can stimulate endocrine (e.g., GLP-1) and neural mechanisms that affect eating and metabolism.

Miller et al. [30] and Alosco et al. [29] studied cognitive performance in patients (mean BMI = 46 kg/m^2^) approved for bariatric surgery; these subjects also suffered from health issues such as hypertension and Type 2 Diabetes mellitus (T2DM). At baseline, subjects scored below standard levels on tests of memory, attention, executive function and language. At 12 weeks and 12 months post-surgery, BMI declined markedly and performance on memory tests was improved, with no change in the subjects who had not had surgery [30]. At 24 months post-surgery, gains in memory and attention were maintained, while executive function continued to improve at 36 months post-surgery. Note that the average loss of ~16 kg of body weight occurred over a period of 24–36 months. Subjects who regained body weight by 36 months showed decreases in attention at that time.

Smith et al. [31] compared the effects of two types of bypass surgery—Roux-en-Y vs. Vertical Sleeve Gastric Bypass (VSGB)—on cognitive function. Subjects who underwent Roux-en-Y surgery lost more weight by 6 months. At 1 year post surgery, subjects with either procedure showed improvement in auditory attention, psychomotor speed and executive function. Body weight loss and cognitive performance were correlated only for the RYGB group, suggesting that factors other than weight loss (e.g., endocrine effects) also influenced cognitive performance following surgery.

These investigations of the effect of CR on cognitive function suggest that: (1) modest energy restriction of healthy, non-obese individuals is unlikely to produce marked changes in cognitive performance [26,28,34]; (2) cognitive function of overweight, older individuals may be positively responsive to a few months of CR [14,25,29,30,31]); (3) dramatic loss of body weight in people with severe obesity, as produced by gastric bypass surgery, may produce lasting improvements in initially impaired cognitive function; (4) dramatic weight loss in a short period of time (very high rate of loss) may be associated with cognitive deterioration [24].

#### 3.1.3. Cognitive Effects of IF vs. CR

Few human trials have directly compared the effects of IF and CR on cognitive outcomes. Two randomized controlled trials that made this comparison and included data on body composition were those of Kim et al. [36] and Teong et al. [37] (Table 3).

The study by Kim et al. [36] evaluated 43 centrally obese (mean BMI~32 kg/m^2^) adults aged 35–75 years who were randomized to either a 5:2 IF regimen (intermittent energy restriction) or daily caloric restriction (CR) for four weeks. Weight loss did not differ between groups. The primary cognitive outcomes were pattern and recognition memory, measured using the Mnemonic Similarity Task, a sensitive proxy for hippocampal neurogenesis. Both groups showed significant improvement in pattern separation scores, with no difference between interventions. However, only the IF group exhibited a significant decline in recognition memory (REC score).

In the study by Teong et al. [37], 48 overweight or obese adults (mean BMI 32.9 kg/m^2^) were randomized to IF (TRF; 16:8) or CR with target energy intakes of 70% of calculated requirements for both groups, in this particular analysis. Body weight and WC decreased in both groups; the reduction in body weight was greater in the IF group. At baseline and at 6 weeks of the study, subjects underwent neuropsychological assessments covering multiple cognitive domains, including executive function, attention, working memory, and verbal learning. Both IF and CR yielded improvements in processing speed and attention, with no difference between groups.

Both studies involved subjects with BMI ~32 kg/m^2^ and both reported many similar effects on cognitive performance by the two approaches, supporting the conclusion that it is the reduction in energy intake and not the pattern of feeding and fasting that leads to changes in cognitive performance. Despite similar weight loss between groups in the study by Kim et al. [36], some decrease in cognitive performance was seen only in the IF condition. This difference between the studies may be due to the higher rate of weight loss in the study by Kim et al., where subjects lost ~3 kg over 4 weeks vs. ~3 kg over 8 weeks in the study by Teong et al. [37]. Periods of greater restriction in an IF protocol may carry potential costs to recognition memory.

Considering all of the human trials reviewed (Table 1, Table 2 and Table 3), there is no clear evidence that the time pattern or the use of periods of fasting themselves affected the cognitive outcome. Instead, the magnitude or rate of weight loss in overweight or obese subjects may produce the cognitive improvements. Furthermore, it may be relevant whether or not subjects are actively losing weight (being in negative energy balance) at the time cognitive testing is performed; individuals after the weight maintenance phase [14], and those who were re-gaining weight [30] showed a decline in cognitive scores.

### 3.2. Question 2: Is Reduction in Obesity (Loss of Body Fat), or Improvement in Insulin Resistance a Necessary Component of the Effect of IF or CR on Cognitive Performance?

Indices related to loss of body fat tracked along with changes in cognitive function in most of the IF and CR trials. In the study by Kim et al. [36], both IF and control groups lost WC, and both showed improvements in cognitive performance; in the paper by Prehn et al. [14] only the CR group lost WC and only CR produced cognitive improvement; in the study by Anton et al. [21] , neither group lost WC and neither showed cognitive improvement; and according to Ooi et al. [22] only the IF group decreased WC and showed increased cognitive scores. However, when a large battery of cognitive tests was performed [15], groups that were similar in loss of WC were only similar in some measures of cognitive performance, with the IF group showing more improvement in some measures and a deterioration in another. Finally, results differed for non-obese subjects in the CALERIE-2 trial, where the CR group lost more body fat but groups did not differ in tests of cognitive function [26,34,35]. These data are consistent with the involvement of loss of body fat, particularly central fat, in the mechanism through which weight loss affects cognitive performance in subjects with obesity.

In those studies in which information related to insulin resistance was supplied, improvements in cognitive performance tracked with improvements in glycemic control or insulin sensitivity. In Witte et al. [25] , only CR lowered fasting insulin levels (an indicator of improved insulin sensitivity) and improved cognitive function; in Prehn et al. [14] CR lowered peak plasma glucose levels during an oral glucose tolerance test, as well as HbA1c, and also improved cognitive performance; in Ooi et al. [22], only the IF group showed reduction in plasma insulin and fasting glucose levels and improvement in cognitive performance. In the CALERIE-2 trial, both groups showed improvement in fasting insulin levels and the related HOMA-IR but neither group showed a change in cognitive performance [26,34,35], suggesting that obesity and its metabolic effects at baseline may be necessary for an effect on cognitive function to occur. In contrast, Kapogiannis et al. [15] reported that, although both IF and control groups showed a reduction in HOMA-IR and neuronal insulin resistance and had similar gains in some cognitive domains, the IF group improved more in other cognitive measures than did those on the control regimen. Taken together, the results are consistent with a role for improved insulin resistance or glycemic control in the enhancement of cognitive function in individuals without diabetes.

## 4. Discussion

Analysis of the trials investigating the effects of IF or CR on cognitive function support the shared mechanism of adequate weight loss in individuals with overweight or obesity. Given that only partial advantage was seen in one study [15], and that others showed no advantage of IF over CR in improving cognitive function, it appears that weight loss and fat loss, however achieved, are the key to cognitive improvement. In addition, most studies supported a relationship between these effects and improvements in insulin resistance or glycemic control (a benefit of improved insulin resistance). It must be appreciated that this analysis cannot go beyond pointing out a relationship among these factors and cannot imply causation. For example, the improvements in insulin resistance may be a separate benefit of the loss of body fat and not be a mediator of the effect on cognitive function.

Visceral adiposity has been reported to play a significant role in the pathophysiology of age-related cognitive decline [12,13,38]. Visceral adipose tissue is an active endocrine organ that secretes proinflammatory cytokines such as tumor necrosis factor-alpha (TNF-α), interleukin-6 (IL-6), and C-reactive protein (CRP), which contribute to systemic low-grade inflammation, a hallmark of obesity-related metabolic dysfunction [39]. These inflammatory mediators disrupt insulin signaling and promote vascular dysfunction, two mechanisms that can compromise brain function over time [40].

Systemic inflammation stimulated by adipose tissue may influence cognitive decline via effects on the blood–brain barrier (BBB) [39,41]. High-fat diets and obesity are associated with impaired BBB integrity, resulting in increased permeability to cytokines and neurotoxins, particularly in regions such as the hippocampus that are critical for memory and learning [39]. Peripheral cytokines such as IL-6 and TNF-α can also activate central immune responses, triggering local cytokine production and exacerbating hippocampal inflammation [42,43,44,45].

Several epidemiological studies show that systemic inflammation is associated with cognitive decline in aging populations, supporting the hypothesis that proinflammatory cytokines may contribute to neurodegenerative processes [39,40,46,47]. In the longitudinal Sydney Memory and Ageing Study, Trollor et al. [40] found that elevated IL-6 levels were significantly associated with slower processing speed and poorer executive function, even after controlling for key confounders including depression, vascular risk factors, and APOE-ε4 status. Yaffe et al. [46] conducted a prospective cohort study involving 3031 well-functioning African American and white subjects enrolled in the Health, Aging, and Body Composition Study. Participants were free of dementia at baseline and underwent repeated cognitive assessments using the Modified Mini-Mental State Examination (3MS). Subjects in the highest tertiles for IL-6 and CRP performed significantly worse on cognitive tests at follow-up than those in the lowest and had higher odds of cognitive decline over two years. Dik et al. [47] examined the association between serum inflammatory proteins and cognitive decline in older Dutch adults without dementia. The study found that higher IL-6 levels were associated with greater cognitive decline across domains such as memory and processing speed over a 3-year period. Despite differences in sample demographics and geographic regions, all three studies reported worse cognitive outcomes in groups with the highest inflammatory burden compared to those with the lowest and support a mechanistic pathway linking central obesity to cognitive decline via systemic and central inflammation. Reduction in central obesity and its attendant inflammatory state may lead to the cognitive improvements achieved by weight loss through IF or CR.

## 5. Conclusions

The analysis in this review supports the conclusion that Intermittent Fasting exerts its effect on cognitive function through negative energy balance, and not through the imposition of a pattern of fasting and feeding independent of weight loss. Achievement of adequate loss of body weight or body fat by adults with overweight or obesity, either through IF or CR, is associated with changes in cognitive performance in clinical trials. Further investigation of the effect of IF on particular domains of cognitive function may be warranted. The metabolic changes produced by the loss of body fat, such as a reduction in inflammation and a diminution of insulin resistance, may contribute to the cognitive improvement observed with intermittent fasting and caloric restriction.

## 6. Limitations

Given the dearth of studies that matched our inclusion and exclusion criteria, there are significant limitations on our analysis and conclusions: (1) the large variety of cognitive tests and cognitive domains that were addressed could not be compared between IF and CR or even within similar protocols, preventing any understanding of which domains might be more amenable to the effects of either IF or CR. (2) Only categorical (yes/no) comparisons were made, instead of correlations between the degree of body weight or body fat loss (for example) and cognitive performance. (3) The various forms of IF (ADF, ADMF, TRF, 5:2) were taken as one category, preventing an analysis of particular forms of IF on cognitive performance.

## Figures and Tables

**Figure 1 nutrients-17-02407-f001:**
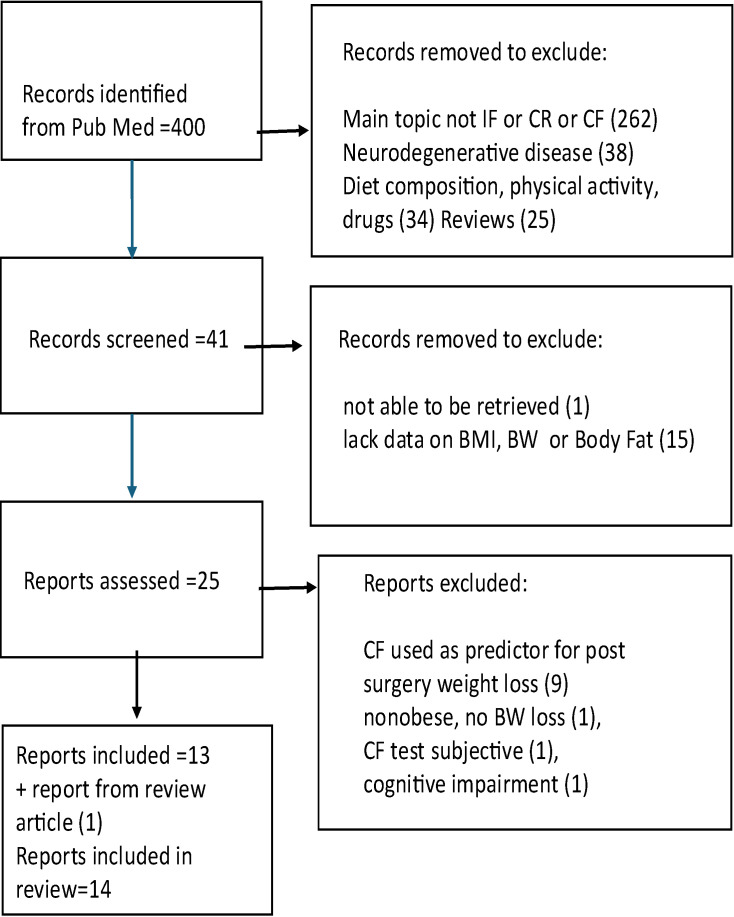
Flow chart of literature search.

**Table 1 nutrients-17-02407-t001:** Effect of intermittent fasting on cognitive function.

Author and Year	Subjects	Intervention	Control Group	CF Measures	Results
**Anton et al.** **(2019)** [21]	≥65 years Men and women With slow gait BMI 25–40 kg/m^2^ *n* = 10	4 weeks IF (16:8 TRE)	Same subjects pre vs. post	Attention, memory, and executive function tests: MoCA	No effect on CF IF: ↓ BW: 97 to 95 kg ↓ **BMI**: 34 to 33 kg/m^2^
**Kapogiannis et al.** **(2024)** [15]	63.2 years Men and women Insulin resistant BMI 35 kg/m^2^ IF, 33.1 kg/m^2^ C *n* = 20 IF 20 C	8 weeks 5:2 IF or HL diet	Healthy Living (HL) diet Not energy reduced	Memory, executive function tests: CVLT, Executive function composite test, Fluency factor, Dimensional set shifting	Both groups ↑ memory. effect IF > C Both ↑ executive function IF > control ↓ BW, ↓ **BMI** −1.41 kg/m^2^ IF > Control **↓ WC** IF = Control IF = Control: ↓ HOMA-IR, ↓ HbA1c ↓ neuronal inflammation
**Ooi et al.** **(2020)** [22]	68.7 years Men and women BMI 24.2–30.0 kg/m^2^ *n* = 37 IF 27 C	36 months IF (Muslim Sunnah fasting, 2 days/week)	Non-fasting group (non-IF)	Memory, executive function, and attention tests: MMSE, MoCA, RAVLT, Digit Span test, Digit Symbol	IF only: ↑ memory, executive function, attention. ↓ **BW**: 52.8 to 49.2 kg ↓ **BMI**: 24.2 to 22.6 kg/m^2^ ↓ **WC**: 86.6 to 81.5 cm ↓ fasting insulin and glucose, triglyceride, LDL, and total cholesterol, MDA, and CRP, and higher HDL

BMI = Body Mass Index; BW = body weight; BW = Body weight; CF = cognitive function; CVLT = California Verbal Learning Task; HDL = High Density Lipoprotein; HL = Healthy living; IR = insulin resistance; LDL = Low Density Lipoprotein; MDA = malodialdehyde; MMSE = Mini-Mental State Examination; MoCA = Montreal Cognitive Assessment; RAVLT = Rey Auditory Verbal Learning Test; WC = waist circumference; ↑—increase; ↓—decrease.

**Table 2 nutrients-17-02407-t002:** Effect of caloric restriction on cognitive function.

Author and Year	Subjects	Intervention	Control Group	CF Measures	Results
**Witte et al.** **(2009)** [25]	60.5 years Men and women BMI 28.2 kg/m^2^ *n* = 20 CR, 10 C	3 months CR (30% energy reduction)	Habitual ad lib intake	Attention and working memory tests: AVLT, TMT A/B WMS-R	CR ↑ working memory, correlated with ↓ fasting insulin and CRP. CR: ↓ **BW** 87.9 to 85.5 kg ↓ **BMI** 29.9 to 29.1 kg/m
**Prehn et al.** **(2016)** [14]	61 years Only women BMI > 27 *n* = 19 CR, 18 C	12 weeks CR + 4 weeks in energy balance	Habitual ad lib intake	Recognition memory and attention tests: VLMT, TMT A/B, Stroop test	CR only ↑ recognition memory improved brain structure. Effects only during CR. CR only: ↓ **BW** 93.2 to 79.7 kg ↓ **BMI** 34.7 to 29.6 kg/m^2^ ↓ **WC** 114.6 to 101.6 cm. Reductions maintained @ 4 weeks in energy balance
**Kretsch et al.** **(1997)** [24]	23–42 years Only women BMI 31.5 kg/m^2^ CR, 34.2 kg/m^2^ C *n* = 14 CR, 11 C	15 weeks 50% CR + 3 weeks of weight stabilization	Habitual ad lib intake	Memory, attention, and reaction time tests: Bakan vigilance task, Word recall task, two finger tapping task, Eriksen effect	CR ↑ short-term memory ↑ reaction time (neg) CR: ↓ **BW** 86.6 to 74.3 kg ↓ **Body Fat**: −10.4 kg
**Martin et al.** **(2007)** [26]	25–50 years Men and women BMI = 25–30 kg/m^2^ *n* = 12 CR 12 C	6 months 25% CR (CALERIE trial)	Habitual ad lib intake	Memory and attention tests: RAVLT, ACT, BVRT, CPT-II	CR no effect on memory or attention. CR only ↓ BW 80.9–10.4% @ 6 months
**Leclerc et al.** **(2020)** [27]	21–50 years Men and women BMI 22–28 kg/m^2^ *n* = 220	2-years 25% CR (CALERIE trial)	Habitual ad lib intake	Working memory tests: CANTAB	CR > Control: ↑ working memory.
**Silver et al.** **(2023)** [28]	38.1 ± 7.2 years Men and women BMI 25.1 kg/m^2^ *n* = 143 CR 75 C	2-years 25% CR (CALERIE trial)	Habitual ad lib intake	Working memory tests: CANTAB	CR no effect on working memory. CR > C baseline
**Alosco et al.** **(2014)** [29]	20–70 years Men and women with MetSyn BMI 46.61 kg/m^2^ *n* = 50	36 and 48 months post bariatric surgery	Same subjects pre- vs. post-surgery	Memory, attention, and executive function tests: Digit span, TMT, Stroop Test, Maze Task, Verbal List-learning	Surgery: ↑ attention (to 24 months; effect lost if BW regain), executive function (peak 36 months) memory (to 36 months), Surgery: ↓ **BMI** 46.6 to 32.4 kg/m^2^
**Miller et al.** **(2013)** [30]	19–61 years Men and women bariatric BMI 46.2 kg/m^2^ *n* = 95, 42 C	12 months post bariatric surgery. Follow-ups at 12 weeks and 12 months	Obese controls BMI 40.77 kg/m^2^	Memory, attention, and executive function tests: Cognitive Test Battery, VLL, Digit Span, VI, Maze Task	Surgery only: ↑ memory and executive function @ 12 months Surgery only: ↓ **BMI** 46.2 to 30.2 kg/m^2^
**Smith et al.** **(2023)** [31]	18–55 years women BMI VSGB = 43.9 kg/m^2^ *n* = 17 BMI RYGB = 44.5 kg/m^2^ *n*= 18	Follow-up at 2, 12, 24, and 52 weeks post- VSGB or RYGB surgery.	Same subjects pre- and post-intervention	Auditory attention, processing speed, memory, and executive function tests: LNS, HVLT, SCWT, SDMT, TMT A and B	RYGB and VSGB: ↑ different domains of executive function, processing speed, and auditory attention. VSGB: ↓ BMI by 10 kg/m^2^@48 weeks RYGB: ↓ BMI by 14 kg/m^2^@48 weeks

Means of data are shown. ACT = Auditory Consonant Trigram; AVLT = Rey Auditory Verbal Learning Task; BVRT = Benton Visual Retention Test; BW = body weight; CANTAB = Cambridge Neuropsychological Tests Automated Battery; CF = cognitive function; CPT-II = Conners’ Continuous Performance Test-II; HVLT = Hopkins Verbal Learning Test; LNS = Letter Number Sequencing test; RAVLT = Rey Auditory Verbal Learning Test; SCWT = Stroop Color and Word Test; SDMT = Symbol Digits Modality Test; TMT A/B = Trail Making Test A and B; VI = Verbal Interference; VLL = Verbal List Learning; VLMT = Verbal Learning and Memory Test; WMS-R = Wechsler Memory Scale-Revised . ↑—increase; ↓—decrease.

**Table 3 nutrients-17-02407-t003:** Effect of IF vs CR on Cognitive Function.

Author and Year	Subjects	IF Group	CR Group	CF Measures	Results
**Kim et al.** **2020** [36]	35–75 years Men and women With central obesity BMI = 32.0 kg/m^2^ IF, 30.9 kg/m^2^ C *n* = 20 IF 23 CR	4 weeks 5:2 Intermittent energy restriction Med-style diet pre v post	500 kcal/d deficit), both on Med-style diet pre v post	Pattern separation and recognition memory tests: Mnemonic Similarity Task	CF: No group difference Improvement in Combined Cohorts Both diets **↓ BMI** IF −3.1 kg/m^2^ CR −2.8 kg/m^2^ ↓ **WC** IF = C
**Teong et al.** **2021** [37]	35–70 years women BMI 25–42 kg/m^2^ *n* = 22 IF 24 CR	8 weeks IF (16:8 TRF) 30% energy reduction Pre v post	8 weeks (30% kcal restricted) pre v post	Attention, processing speed tests: DSST, PVT	↑ CF IF = CR ↓ **BMI**: CR −1.4, IF −2.0 kg/m^2^ Effect IF > CR **↓ BW** CR −3.9, IF −5.4 kg) Effect IF > CR ↓ **BF** CR −2.8, IF −3.9 kg Effect IF > CR

BF = Body fat; BMI = Body Mass Index; BW = Body weight; CF = cognitive function; DSST = Digit Symbol Substitution Test; PVT = Psychomotor Vigilance Task. ↑—increase; ↓—decrease.

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
