# Peer review of "Does Energy Restriction and Loss of Body Fat Account for the Effect of Intermittent Fasting on Cognitive Function?"

_nutrients, 2025, doi:10.3390/nu17152407_

Round 1
Reviewer 1 Report
Comments and Suggestions for Authors
This review presents an interesting question regarding two diet strategies, intermittent fasting and caloric restriction, and improvements in cognition, asking is it the diet strategy or the diet-induced weight loss contributing to improved cognition. To my knowledge, this is a novel idea that will contribute new knowledge to the literature. The review does not follow the PRISMA guidelines, which is unfortunate as this leaves the thoroughness/transparency of the review in question. At minimum, the authors need to include a flow chart of the literature search outlining number of studies identified and study selection.
Front matter: full names need to be listed. CR needs to be in the title.
Backmatter is missing
Page 4: it is unclear what “Sharifi” signifies
Page 6: The first sentence of the final paragraph needs to be rewritten for clarity – what is the %CR for the Prehn et.al. study?
Page 8: The last sentence of the second paragraph discussing spatial working memory contradicts what is stated previously (e.g. that ‘no differences in cognitive function were seen between’ groups). Please clarify and provide more detail regarding the dispute of the data (ref. 31).
Page 11: delete the last sentence of the 1st paragraph.
The conclusion must focus on the research questions and include statements for both IF and CR. The statements regarding IF ‘success’ are not factual, particularly regarding potential social and lifestyle disruptions.
Author Response
Page 4: it is unclear what “Sharifi” signifies
Response:
We have inserted the reference number and removed the name, to correct the error
3.Page 6: The first sentence of the final paragraph needs to be rewritten for clarity – what is the %CR for the Prehn et.al. study?
Response:
We have added the information about the energy restriction in the CR condition:
“8wk@800kcal/d, 4wk@unspecified energy reduction until >10% loss of body weight,4 wk of weight maintenance”.
4.Page 8: The last sentence of the second paragraph discussing spatial working memory contradicts what is stated previously (e.g. that ‘no differences in cognitive function were seen between’ groups). Please clarify and provide more detail regarding the dispute of the data (ref. 31).
Response:
We have added information concerning the opposing findings of Leclerq et al. and Silver et al. With the information given in the two papers it is not clear how they can be reconciled. They appear to be referring to the same subjects, although in Leclerq N= 220 and in Silver N=218. As now noted in the text, only Silver et al. reported that the CR group had fewer errors at baseline (better performance) than the ad lib control group. Perhaps differences in analytical approach could be relevant. Leclerc et al. classified their finding as a “slightly positive effect”.
“Leclerq et al. (ref) reported a “slightly positive effect” on spatial working memory (SWM) after 24 months only for the CR group, based on their total SWM errors but not SWM strategy. However, utilizing the same available data set, Silver et al. (ref) concluded that there was no difference in effect between the CR and control subjects; ony Silver et al. reported that the CR group performed significantly better at baseline than the control group.”
5.Page 11: delete the last sentence of the 1st paragraph.
Response;
We have deleted the sentence: “The next question concerns what mechanisms may be likely to connect reduction in body fat and insulin resistance with improved cognitive performance.”
6.The conclusion must focus on the research questions and include statements for both IF and CR. The statements regarding IF ‘success’ are not factual, particularly regarding potential social and lifestyle disruptions.
Response
CONCLUSION
The analysis in this review supports the conclusion that Intermittent Fasting exerts its effect on cognitive function through negative energy balance, and not through the imposition of a pattern of fasting and feeding independent of weight loss. Achievement of adequate loss of body weight or body fat by adults with overweight or obesity, either through either IF or CR, is associated with changes in cognitive performance in clinical trials. Further investigation of the effect of IF on particular aspects of cognitive function may be warranted.Formatting...The metabolic changes produced by the loss of body fat, such as reduction of inflammation and diminution of insulin resistance, may contribute to the cognitive improvement observed with intermittent fasting and caloric restriction.
I could not continue my response to the Reviewer using the format provided on this site; I could not re-locate it.
Reviewer 2 Report
Comments and Suggestions for Authors
Dear authors,
Thanks for your submission.
The study is interesting but very limited.
- A "study selection flowchart" should be inserted, highlighting the number of studies which were excluded and how the studies were selected for analysis by the authors.
- Insert "Methodological Quality of the Study"
- A "PICO" table should be created.
- The tables are confusing; the descriptions should be clearer. The locations where the studies were conducted should be added.
Author Response
A "study selection flowchart" should be inserted, highlighting the number of studies which were excluded and how the studies were selected for analysis by the authors.
Response: We have inserted the flow chart as Figure 1 in the manuscript.
Reviewer 3 Report
Comments and Suggestions for Authors
Ref.: nutrients-3746504
This review is dealing with the mechanism by which intermittent fasting exerts its beneficial effect on cognitive function. Results from human studies, support the notion that this beneficial effect results from reduction of body weight, visceral adiposity and improvement insulin resistance due to negative energy balance, rather than fasting/eating patterns per se.
The review is well-designed and conducted and the 3 tables help to address all questions thoroughly. The discussion is comprehensive and includes not only a critical review of the studies but, in addition, the biochemical mechanisms possibly involved (genetic, inflammatory). Limitations are presented and suggestions for future studies are included in the discussion.
I believe that the present review adds to the current literature and it will receive many citations.
One question: Are the names of the authors written correctly?
Author Response
Are the names of the authors written correctly?
Response
Thank you for noticing the error. We have corrected the authors' names. Thank you for your review.
Round 2
Reviewer 1 Report
Comments and Suggestions for Authors
Your attention to my concerns is appreciated. Good luck with your paper.